# The Microbial Mechanisms of a Novel Photosensitive Material (Treated Rape Pollen) in Anti-Biofilm Process under Marine Environment

**DOI:** 10.3390/ijms23073837

**Published:** 2022-03-30

**Authors:** Qing-Chao Li, Bo Wang, Yan-Hua Zeng, Zhong-Hua Cai, Jin Zhou

**Affiliations:** 1Shenzhen Public Platform for Screening and Application of Marine Microbial Resources, Institute for Ocean Engineering, Shenzhen International Graduate School, Tsinghua University, Shenzhen 518055, China; 18351004704@163.com (Q.-C.L.); zengyanhua@hainanu.edu.cn (Y.-H.Z.); caizh@sz.tsinghua.edu.cn (Z.-H.C.); 2CAS Key Laboratory of Quantitative Engineering Biology, Shenzhen Institute of Synthetic Biology, Shenzhen Institutes of Advanced Technology, Chinese Academy of Sciences, Shenzhen 518055, China; bo.wang@siat.ac.cn

**Keywords:** rape pollen, photosensitive material, marine antifouling, microbial mechanisms

## Abstract

Marine biofouling is a worldwide problem in coastal areas and affects the maritime industry primarily by attachment of fouling organisms to solid immersed surfaces. Biofilm formation by microbes is the main cause of biofouling. Currently, application of antibacterial materials is an important strategy for preventing bacterial colonization and biofilm formation. A natural three-dimensional carbon skeleton material, TRP (treated rape pollen), attracted our attention owing to its visible-light-driven photocatalytic disinfection property. Based on this, we hypothesized that TRP, which is eco-friendly, would show antifouling performance and could be used for marine antifouling. We then assessed its physiochemical characteristics, oxidant potential, and antifouling ability. The results showed that TRP had excellent photosensitivity and oxidant ability, as well as strong anti-bacterial colonization capability under light-driven conditions. Confocal laser scanning microscopy showed that TRP could disperse pre-established biofilms on stainless steel surfaces in natural seawater. The biodiversity and taxonomic composition of biofilms were significantly altered by TRP (*p* < 0.05). Moreover, metagenomics analysis showed that functional classes involved in the antioxidant system, environmental stress, glucose–lipid metabolism, and membrane-associated functions were changed after TRP exposure. Co-occurrence model analysis further revealed that TRP markedly increased the complexity of the biofilm microbial network under light irradiation. Taken together, these results demonstrate that TRP with light irradiation can inhibit bacterial colonization and prevent initial biofilm formation. Thus, TRP is a potential nature-based green material for marine antifouling.

## 1. Introduction

Biofilms represent a major problem in the worldwide marine industry as they can directly lead to microbial-induced biofouling [1] and result in corrosion of marine monitoring instruments, submarine pipelines, coastal structures, ships, and even sea-crossing bridges [2,3,4]. In addition, the attached microorganisms are transported to other sea areas as ships sail, which may affect the local ecological balance of those areas and cause invasion of alien species [5]. The biofouling phenomenon also results in huge economic losses and a wide range of ecological issues all over the world [6]. Therefore, research has increasingly focused on strategies to inhibit biofouling. The attachment of fouling organisms significantly relies on biofilms; therefore, biofilm formation inhibition is an important antifouling mechanism [7,8].

In the search for a highly efficient, easily processable, low-cost antifouling strategy, research has been performed from many perspectives, including material science and ecology, in recent years [9,10]. However, traditional marine antifouling techniques, which include bactericide coatings, antifouling paints, bionic material, and artificial membrane construction, have one or more unavoidable drawbacks, including recolonization, unacceptably high cost, and secondary pollution [11,12,13,14,15]. Therefore, there is an urgent need to develop novel environmentally friendly and non-toxic marine biofilm inhibitors. Biochemical remedies have been widely applied in biofouling treatment owing to their environmental friendliness, substitutability, sustainability, and safety [16,17]. Novel combined biological-chemical antifouling methods such as the application of photocatalytic materials have also been extensively investigated, as they represent promising “green” strategies [18,19] for developing highly efficient and low-cost antifouling agents [20].

Among the various potential photocatalytic materials, rape pollen (RP), a plant-based material (mainly consisting of a carbon fiber skeleton), has been reported to have a high specific surface area and to show an excellent inhibitory effect on the growth and development of pathogenic microorganisms [21]. Previous studies showed that RP has excellent quantum efficiency and photocatalytic performance, favoring the formation of photogenerated holes and electrons via reducing their diffusion length [22,23,24]. Our previous work showed that treated RP (TRP) could significantly inhibit the growth of *Pseudomonas aeruginosa* and *Escherichia coli* cells under visible light (VL) irradiation [25]. This indicated that TRP may have photocatalytic antifouling capacity. The physical and chemical properties of TRP and its inhibitory effects on single model bacteria have been studied [26]; however, the biological mechanisms underlying its effects on naturally occurring multiple-species communities have remained unknown. This motivated us to explore the role of TRP in the formation of biofilms from mixed natural microbial communities.

To evaluate the antifouling efficacy of TRP in a mimic marine environment, we designed a series of aquariums and hung stainless-steel slices in seawater. The composition and structure of the biofouling microbiome under TRP interference were explored. In addition, the basic features of TRP, the morphology of the biofilm, the development of the biofilm community structure, a possible interaction network, and metagenomic profiles were investigated. This study is intended to supply natural-scale evidence of the anti-biofilm effect of TRP, as well as the mechanisms underlying this effect, thereby shedding light on antifouling mechanisms and potential strategies for developing antifouling agents from natural photocatalytic resources.

## 2. Results

### 2.1. Characterization of TRP

Scanning electron microscopy (SEM) images showed that TRP had a porous three-dimensional network structure with an average width and length of approximately 10 and 20 nm, respectively (Figure 1a). Through ultrasonic peeling treatment, transmission electron microscopy (TEM) images were obtained; these further verified the hollow structure of TRP (Figure 1b). A wide diffraction peak at 21.3° appeared in the X-ray diffraction (XRD) pattern (Figure 1c). The specific surface area of TRP was 14.16 m^2^g^−1^, and the majority of pores were about 40–80 nm in diameter (Appendix A). Through ultraviolet-VL diffuse reflectance spectroscopy (UV-vis DRS) the bandgap of TRP was calculated to be about 1.53 eV (Figure 1d). Energy-dispersive X-ray (EDX) spectroscopy showed that TRP was composed of C, O, S, and N elements at percentages of 55.56%, 28.61%, 10.75%, and 5.08%, respectively (Figure 1e). The Fourier transform infrared (FT-IR) spectrum (Appendix A) showed that TRP was mainly composed of carbon-skeleton-based components (C–O, –CH_2_–, and –CH=CH–). 

To further evaluate the potential oxidation activity of TRP, electron spin-resonance (ESR) analysis was performed. As shown in Figure 1f,g, from pristine TRP to ageing (28 days) TRP, both DMPO-•O_2_^−^ (superoxide ion) and DMPO-•OH (hydroxyl radical) signals were detected under VL irradiation, whereas no •O_2_^−^ or •OH signal was detected in the dark.

### 2.2. Effect of TRP on the Biodiversity of Biofilm

The formation of a surface biofilm was observed in the control groups (P and L) from the first week, and it continued to grow until the fourth week (Figure 2). The control groups accumulated greater levels of filming under the seawater condition. When the slices were exposed to TRP under VL irradiation (group PL), biofilm development was inhibited.

Confocal laser scanning microscopy (CLSM) three-dimensional images were used to precisely characterize the biofilm structures at different time-points (Figure 2). For example, during the first week of treatment, biofilms formed in the control groups (P and L) were denser and thicker with more cells than those in the test group (PL). In the control group, the bacteria attached to the flakes to form biological films. By contrast, in the test group (PL), the biofilms were scattered, showing a disrupted surface topology pattern. Moreover, in the early, middle, and late stages of biofilm formation, the highest dead cell ratio was observed in the PL group, indicating that TRP indeed had an inhibitory effect on biofilm formation. It is noteworthy that TRP alone (group P) was only moderately effective against biofilm, indicating that the biological role also relies on light catalyzation. In the CLSM images, the phenomenon of biofilm aging was obvious in the PL group, which showed a relatively high percentage of dead cells in the late stage. After quantification of biofilm parameters by COMSTAT analysis (taking the second week as an example), we found that the test group (PL) inhibited biofilm total biomass and average thickness by 2.05-fold and 1.34-fold, respectively, as well as increasing the roughness coefficient (Figure 3a–c).

### 2.3. Effect of TRP on the Biodiversity of Biofilm

The rarefaction and alpha- and beta-index values of the microbial communities were calculated to estimate the diversity of microorganisms in seawater and in different biofilms. The rarefaction curves reached a plateau in all tested samples, indicating that the number of sequences identified was sufficient to describe biofilm microflora (Appendix A). Alpha-diversity analyses showed that the richness, diversity, and uniformity of the PL group were lower than those of other biofilm groups in the first and second weeks (*p* = 0.002 in the first week and *p* = 0.0009 in the second week) (Figure 4a–c). However, the alpha-diversity of the PL group gradually increased over time and did not display any significant difference when compared with other groups in the fourth week (*p* > 0.05).

For beta-diversity analysis, the separation patterns between seawater and four different groups were identified using nonmetric multidimensional scaling (NMDS) based on the Bray–Curtis dissimilarity matrix. A clear separation (permutational multivariate analyses of variance (PERMANOVA), R^2^ = 0.402 to 0.642, *p* < 0.05) based on the operational taxonomic unit (OTU) composition was found in the biofilm microbial community structure at three timepoints, whereas microbial community structures in seawater clustered together (Figure 4d). As shown in Appendix A, the bacterial communities of the biofilm in the PL group clustered away from other groups in the initial stage (first week). However, the separation of biofilm communities did not exhibit a significant difference among all groups in the later stage (fourth week), indicating that the effect of TRP on β-diversity is not significant in mature stages of biofilm development.

### 2.4. Taxonomic Composition

A total of 27 phyla were identified based on the classification of 2,900,000 16S ribosomal RNA (rRNA) gene sequences; 1186 OTUs per biofilm sample and 1313 OTUs per seawater sample were detected on average (Appendix A).

Taxonomic analysis of the 16S rRNA genes identified the top 20 phyla across all biofilm samples (Figure 5a), of which Proteobacteria were predominant (representing more than 80%). Bacteroidetes, Actinobacteria, and Cyanobacteria showed high abundance (up to 15%) in the seawater group; however, this declined to less than 2% in the biofilm-related groups. At the genus level, univariate analysis was conducted on representative taxa to show the changes more clearly. As shown in Figure 5b, *Alteromonas* and *Oceanicaulis* were more abundant (>three-fold) in the biofilm microbiomes of the test group (PL) compared with the control group. In addition, although the overall proportions of *Parvularcula* and *Maricaulis* were relatively low in the test group (PL) in the first week, they had increased more than 40-fold (from 0.18% and 0.13% to 9.01% and 5.26%, respectively) by the fourth week.

Supplementary evidence was obtained from the linear discriminant analysis effect size (LEfSe) (LDA score > 4, from phylum to genus) results, which showed that the type and number of biomarkers (keystone species that indicate changes in biofilm structure) in the PL group were obviously different from those of the seawater, control (P and L), and blank (N) groups. As shown in Appendix A, the majority of biomarkers in seawater microbial communities did not change during the entire period of cultivation, whereas the biomarkers in biofilms varied between different periods. The main biomarkers in the test group were *Oceanicaulis*, *Hyphomonadaceae*, *Alteromonas*, *Alteromonadaceae*, *Pseudoalteromonas*, *Pseudoalteromo-nadaceae*, and *Vibrio* in the first week, changing to *Oceanicaulis*, *Hyphomonadaceae*, and *Vibrionaceae* in the second week, and *Parvularculaceae*, *Limnobacter*, and *Halomonadaceae* in the fourth week. The number of biomarkers in the test group decreased with increasing incubation time, whereas no significant changes were observed in the control and blank groups.

### 2.5. Correlation Analysis of Environmental Factors

To explore whether environmental parameters led to biofilm suppression, a series of environmental parameters including pH, salinity, temperature, nutrients (total organic carbon [TOC], NH_4_^+^-N, NO_2_^−^-N, NO_3_^−^-N, and PO_4_^3−^-P), and chlorophyll a were examined (Appendix A). The results indicated that the ranges for pH, salinity, temperature, and chlorophyll a were 8.60–8.74, 31.91–33.78%, 27.63–29.09 °C, and 0.15–0.20 µg/L, respectively. The ranges for TOC, PO_4_^3−^-P, NO_3_^-^-N, NO_2_^-^-N, and NH_4_^+^-N were 1.84–2.08 mg/L, 0.020–0.035 mg/L, 0.018–0.060 mg/L, 0.002–0.010 mg/L, and 0.142–0.216 mg/L, respectively. In addition, redundancy analysis (RDA) was conducted to explore the effects of environmental factors on bacterial community structures of biofilms. As shown in Figure 5c, the diversity of the bacterial communities was positively correlated with temperature, NO_2_^−^-N, PO_4_^3−^-P, and pH but negatively correlated with TOC, NH_4_^+^-N, chlorophyll a, and salinity. In summary, the explanatory variables accounted for approximately 20.6%. 

### 2.6. Network Analysis of Biofilm Microorganisms

To investigate the relationships between taxa in the bacterial communities, pairwise Spearman correlation coefficients were calculated to detect the co-occurrence patterns of significant taxa (ρ ≥ 0.7, *p* < 0.05, top 50). As shown in Figure 6, the percentage of positive correlations in the test group (PL) was higher than that in the control groups (P and L), blank group (N), and seawater group. Moreover, the network structure of the microbial communities of the test group had the highest number of edges, demonstrating that the bacterial community network of the test group was more complex than that of the other groups.

In order to describe the topology of the network, the average degree, average path length, average clustering coefficient, modularity, and figure density were calculated. As shown in Table 1, the number of edges, average degree, average clustering coefficient, and figure density were higher in the test group (PL) than in other groups. The modularity index values were 0.281, 0.468, 0.346, 0.436, and 0.398 in the test group (PL), control group (P), control group (L), blank group (N), and seawater group, respectively. In addition, when the modularity index was >0.4, the network had a modular structure and could be divided into several independent groups, indicating that there was a tighter network structure in the test group compared with the other groups. These results show that TRP, especially with light irradiation, can increase co-occurrence patterns and network complexity, thereby influencing the environmental buffering capacity and stability of the microbial communities of biofilms. Additional evidence was obtained from the keystone species in each group (Appendix A), which have fundamental roles in maintaining the ecological networks of microbial communities [27,28].

### 2.7. Biofilm Microbial Functions based on Metagenomic Data

To investigate the overall effect of TRP on biofilm function, we performed metagenomic analysis to evaluate its metabolic potential. Approximately 95 Gb metagenomic DNA sequences (150-bp read length and paired-end sequencing) were obtained for eight samples. Read information for the high-throughput sequencing data is shown in Appendix A. In total, 7755 unique Kyoto Encyclopedia of Genes and Genomes (KEGG) protein families were identified from the eight metagenomes. From the bird-view, 428 metabolic pathways were classified in the metagenome data based on the level 3 category.

We next explored whether TRP could simultaneously induce changes in community structure and community function. From the level 2 category (Figure 7a), genes involved in environmental adaptation, cellular community, drug resistance, signal transduction, cell motility, translation, replication and repair, glycan biosynthesis, and metabolism were enriched in the PL group, indicating that processes related to environmental tolerance, the cell aging process, and cell–cell communication were more active. However, genes involved in carbohydrate metabolism, membrane transport, and xenobiotics biodegradation were prominent in the L group.

In the KO category, the metabolic potential of bacterial communities exhibited differences among various groups. The top 30 most abundant genes involved in quorum sensing (QS), biofilm formation, ABC transporter, and oxidation resistance are shown in the heatmap in Figure 7b. Most of the genes related to ABC-2 type transport system permease protein (ABC-2.P), putative ABC transport system permease protein (ABC.CD.P), ABC-2 type transport system ATP-binding protein (ABC-2.A), methyl-accepting chemotaxis protein (mcp), two-component system, biopolymer transport protein (exbB), chemotaxis protein (cheY), flagellin (fliC), and acyl-CoA thioester hydrolase (ybgC) were greatly enriched in the PL group. Conversely, genes related to protein SCO1, transposase, ParB family transcriptional regulator (parB), and putative transposase were downregulated in the test samples (PL group). A more intuitive picture (based on RDA) of the correlations between functional genes and groups is given in Appendix A.

### 2.8. Linkage of TRP to Biofilm Inhibition

The effects of microbial communities, microbial networks, environmental factors, and gene alterations (downregulation or upregulation) on biofilm development were evaluated using the structural equation model (SEM) (Figure 8). Chemical environmental factors, upregulation of genes, the microbial network, and microbial communities significantly (*p* < 0.01) affected biofilm development. Their standardized influencing factors with respect to the biofilm were 1.046, 0.543, 0.314, 0.242, and 0.216, respectively.

## 3. Discussion

### 3.1. Material Characteristics

The SEM images (Figure 1a) indicated that TRP partially retained the morphology and structure of RP. Moreover, abundant pore channels were observed, implying that TRP may have strong light-trapping ability. TEM images obtained through ultrasonic peeling treatment further verified the hollow structure of TRP (Figure 1b). A wide diffraction peak at about 21.3° in the XRD pattern was observed, indicative of the amorphous nature of TRP (Figure 1c). The specific surface area of TRP was 14.16 m^2^g^−1^ and the majority of pores in TRP were about 40–80 nm, indicating a large surface area and volume that would be conducive to light absorption (Appendix A).

To explore the light absorption capacity of TRP, the optical properties of TRP were studied using UV-vis DRS measurements (Figure 1d). The bandgap of TRP was about 1.53 eV, indicating that TRP has a strong light absorption capability ranging from UV to near-IR. EDX spectroscopy showed that TRP was composed of C, O, S, and N elements at percentages of 55.56, 28.61, 10.75, and 5.08, respectively (Figure 1e). The FT-IR spectrum (Appendix A) showed that TRP was mainly composed of carbon-skeleton-based components (C–O, –CH_2_–, –CH=CH–), which confirmed the metal-free character of TRP. ESR analysis was performed to evaluate the potential oxidation activity of TRP. As shown in Figure 1 (f and g), both DMPO-•O_2_^−^ and DMPO-•OH signals were detected under VL irradiation, whereas no •O_2_^−^ and •OH signals were detected in the dark, suggesting that the oxidative ability of the material is light dependent. Furthermore, the oxidative properties of TRP were not significantly weakened after 28 days of treatment, suggesting that its oxidant capacity would have good persistence in the natural environment.

Taken together, our results showed that a “green” porous photocatalytic material was successfully prepared based on a natural substance. Its characteristics make RP an ideal light-active material. Our previous study demonstrated that RP possessed a strong broad-spectrum photocatalytic inactivation capability towards numerous waterborne bacteria in indoor conditions [22,25]. Therefore, TRP could be applied in more fields, such as for antifouling in the natural marine environment.

### 3.2. Anti-Biofilm Profiles of TRP

The CLSM results (Figure 2 and Figure 3) showed that the proportion of dead cells in the PL group was higher than that in the control groups. The underlying mechanism may be related to cell membrane damage. Song reported that live bacteria emit green fluorescence while those possessing a damaged membrane emit red fluorescence [29]. Similarly, in this work, we observed that in the early stages, the cell membranes were not affected much and exhibited relatively strong green fluorescence (Figure 2). However, with prolonged photocatalytic treatment, the red fluorescence signal increased, indicative of membrane damage during the process of photocatalytic inactivation.

The microbial diversity of biofilm under TRP exposure was also analyzed, and a significant difference was observed in α-diversity in the first 2 weeks (*p* < 0.05). Lower Chao 1, Shannon index, and Simpson index values were observed in the PL group compared with the control groups (P and L) (Figure 4a–c). This could be explained by the antibacterial activity of the TRP, which inhibited the colonization of free-living bacteria and affected the richness index. Similarly, beta-diversity alteration was found according to the NMDS results (Figure 4d), indicating the potential separation of the treatment group (PL) from the control and blank groups (P, L and N) as early as the first week. These results suggest that TRP can change the biodiversity of a symbiotic community, alter the relative richness of bacteria, and disturb the normal flora, thereby suppressing biofilm formation [30]. Notably, the microbial communities of biofilms under the four different conditions were clearly separated in the initial stage (first week), whereas some overlap appeared between treatment and control groups in the final stage (fourth week) (Appendix A). This indicates that TRP had stronger biological activity in the early stage than in the mature stage. Supplementary evidence was provided by the PERMANOVA results (Appendix A).

As shown in Figure 5a,b, substantial differences were observed between the control and TRP groups in the microbial composition of biofilm across the whole experimental period. Previous reports showed that Gammaproteobacteria, Alphaproteobacteria, Firmicutes, and Bacteroidetes were the dominant classes in marine biofilms formed on different types of immersed artificial substrates [31], consistent with our results (Figure 5a), i.e., the biofilm in early stage was dominated by Proteobacteria at the phylum level (especially *Alteromonas*). Many genera in Proteobacteria, such as the *Roseobacter* clade and *Alteromonas*, are pioneer colonizers in the aquatic environment owing to their quick responses to organic substances [32,33,34,35]. They supply nutrients and niches for secondary colonizers and therefore have crucial roles in the early stage of biofilm formation [36]. The pioneer colonizers adhere to substrata and excrete valuable metabolic intermediates that contribute to biofilm formation under tough conditions [37]. The relatively high abundance of the above-mentioned species indicates that TRP can alter the species populations and the composition of the biofilm community.

To better explore the influence of TRP on ecological interactions among the multispecies communities of biofilms, and to identify critical indicator taxa [38], co-occurrence networks were constructed, and their topological properties were analyzed (Figure 6, Table 1). The network for the PL group contained 448 edges, a much greater number than that in the other biofilm groups and the seawater group, indicating that the PL group had a higher number of co-occurrences (edge/node ratio = 5.08). In addition, the high average degree and figure density further indicated the greater complexity of the network in the PL group. The network’s complexity is important for ecological buffer capability and microbial homeostasis [39,40,41]. Previous reports showed that environmental pressure could increase network complexity in the phycosphere environment [42]. Under these conditions, in order to obtain more survival opportunities, microorganisms strengthen their interactions (including information exchange, nutrition acquisition, and co-operative behaviors) so the co-occurrence mode will become more diverse [43,44]. Virtually, we observed a higher percentage of positive relationships in the PL group, indicating that the mutual synergistic promotion of microorganisms was tight [45,46]. This phenomenon could be attributed to the selectivity resulting from extreme external pressure [47]. These results indicate that TRP can affect microbial network complexity and the development pattern of biofilms [48].

The degree value of a network provides a local quantification of direct co-occurrence interactions for a specific node, and the betweenness centrality represents the control potential that an individual node exerts over the interactions of other nodes in the network [49,50]. Therefore, species with a low betweenness centrality and high degree were identified as crucial indicator taxa (Appendix A). The keystone species in seawater group were totally different from those in biofilm groups, consistent with the LEfSe analysis results (Appendix A). Among the main keystone species in the PL group, *Maricaulis* and *Labrenzia* were not found in the other three biofilm groups, indicating that they may have important roles under stress conditions. *Maricaulis* can produce exopolysaccharides (EPS) with bio-flocculation activity, providing conditions for growth and development of biofilms under stress conditions [51]. *Labrenzia* also has important effects in biofilm formation and stress resistance [52]. Taken together, the network results suggest that TRP can change the network topological profile in biofilm–keystone species, leading to interference of biofilm formation.

### 3.3. Anti-Biofilm Mechanism at Functional Gene Level

Zooming in on the functional picture showed that key functional pathways and genes associated with cell–cell communication, environmental adaption, and signal transduction were enriched in the PL group (Figure 7). Chemotaxis was the most significant of the functional groups detected. Motility- and chemotaxis-related genes K03406 (*mcp*) and K02406 (*fliC*) were enhanced more than two-fold in the PL group compared with the control groups, possibly owing to environmental pressures during incubation. It has been reported that the enhancement of motile activity is related to overexpression of *fliC* [53]. Our results further demonstrate that TRP has the same properties as another photocatalytic material-Zn^2+^ leached from nano-coating, which can affect bacterial pili and growth under pressure conditions [54]. Chemotaxis is involved in construction of symbiotic relationships and functions in shaping the microbial composition of biofilm [55]. Studies have showed that *fliC* contributes to bacterial swarming motility and biofilm formation [56,57], and lack of *mcp* attenuated the ability of *Cronobacter sakazakii* to form biofilms [58], which is inconsistent with our results. This phenomenon can probably be attributed to the response of the microbial community to extreme conditions through upregulation of motility and chemotaxis genes. The expression of these motility and chemotaxis genes could also be regulated by other biological or environmental factors [59].

It is also worth mentioning that ABC-2.A, which is directly related to QS in the KEGG database, was enriched in the PL group. Research has shown that QS-associated transport systems have critical roles in the stationary phase of survival and growth in bacteria under stress conditions such as oxidative stress, salinity, UV radiation, and starvation [60,61,62]. Several steps in the process of biofilm formation are regulated by QS systems, including primary attachment, cell–cell adhesion, and accumulation [39]. Our results showed that QS genes were more highly expressed in the TRP group compared with the control groups, implying that the QS system exhibits an obvious response under stress conditions. K03566 (*gcvA*) is a glycine cleavage system transcriptional activator and *LysR* family transcriptional regulator. It exhibited a relative high abundance in all experimental groups in our study. These results suggest that TRP treatment with light induced significant changes in biofilm energy and metabolism. Querying the KEGG database showed that *gcvA* was related to the formation of biofilms. Studies have shown that *LysR*-type transcriptional regulators also regulate a series of genes involved in motility, QS, metabolism, and virulence [63,64,65,66]. Concurrently, significantly higher expression of biopolymer transport protein (*ExbB*) (which participates in EPS degradation) was observed in the PL group compared with the control groups (P and L), which may explain the decrease in polysaccharide matrix in biofilm after TRP treatment [29].

The quinone reductase gene (K00344, *qor*), which plays important parts in biological activation and detoxification processes [67], was detected in the four biofilm groups. The abundance of K00344 was higher in the PL group than in the P and N groups (Figure 7b). As shown in Figure 1, many active free radicals such as •O_2_^−^ and •OH were found in the PL group. Therefore, the high abundance of *qor* was conducive to the production of quinone oxidoreductase (NQO1), reflecting the response strategies of antioxidant genes under stress conditions [68]. Imlay reported that environmental and endogenous sources are the main sources of reactive oxygen species (ROS) [69]. Some algal species produce ROS, such as peroxides and hydroxyl radicals, to protect themselves from biofouling [70,71]. In this study, we found a higher proportion of *Oceanicaulis*, which is a cysteine-producer (cysteine has an antioxidant effect) [72,73] in the PL group (Figure 5b). Hence, *Oceanicaulis* was enriched in the PL group owing to the ability of TRP to produce various radicals (•O_2_^−^ and •OH). It can be speculated that the effect of TRP on biofilm microorganisms is exerted mainly through ROS pathway based on the relatively sensitive biological stress response compared with other metabolic capacities. Upon irradiation, photocatalytic materials generate numerous electron–hole pairs, inducing generation of •O_2_^−^, hydrogen peroxide, and •OH [74]. ROS can damage microbial cell membranes, enhance oxidative stress, and induce cell death [75]. In this study, significant VL-driven photocatalytic ability and oxygen stress gene responses were observed in the TRP group, supplying evidence of the ability of TRP to generate ROS. Therefore, the antifouling mechanism of TRP (as a photocatalytic material) is endogenous, and its antifouling effect is environmentally friendly.

According to the overall changes in the metagenomic data, the changes in microbial metabolism induced by TRP were related to functional homeostasis. This provides insight into the antifouling mechanism of TRP. As shown in Figure 8, the biofilm development index was significantly (*p* < 0.01) influenced by upregulation of genes. Significant alterations in oxygen stress and metabolic processes were observed in the PL group, indicating that TRP induced an antioxidation/oxidation imbalance and homeostasis-related metabolic disorder. This is consistent with a previous report stating that biofilm dysbiosis is correlated with metabolism changes via oxidative stress processes [22,25]; ROS generated by TRP were found to destroy the self-protection system of cells. ROS primarily targets the cell membrane and destroys its integrity and functions (e.g., cell respiration and metabolism). In our study, related functional genes were altered in the PL group, which supports the viewpoint mentioned above.

## 4. Materials and Methods

### 4.1. Preparation and Characterization of TRP

TRP was prepared as previously described [25]. Briefly, 50 g RP was added to 500 mL ethanol and sonicated for 48 h until completely dissolved; then, deionized (DI) water was used to wash the filter (TRP was on the filter at the time) three times to improve TRP purity. To retain the morphology of the carbon skeleton, the RP was fixed in 500 mL formaldehyde and ethanol solution (1:1 (*v/v*)) for 12 h with stirring, followed by filtration and washing again. The fixed RP was treated with 500 mL H_2_SO_4_ solution (12 M) for 8 h at 80 °C with stirring. The treated RP was washed with DI water, vacuum-dried overnight, and then ground into powder for subsequent use.

The characteristics of TRP were detected as previously described [25]. Briefly, the surface morphology and element composition of TRP were detected using TEM ( Tecnai G2 f20 s-twin, FEI, USA), field-emission SEM (Zeiss Sigma 300, CarlZeissJena, Germany), and energy-dispersive X-ray spectrometry (Bruker, Germany). The surface area was measured using a porosity analyzer (ASAP2020, Micromeritics, USA), and the pore size distribution was examined using the Barrett–Joyner–Halenda method. The dominant radicals produced by TRP were detected by ESR analysis (JEs-FA300, JEOL, Japan). The micro-crystal structure of TRP was analyzed using XRD (D8 Advance, Bruker, Germany). FT-IR spectra were measured using a FT-IR spectrometer (Frontier, PerkinElmer, Waltham, MA, USA). UV-vis DRS spectra were measured using a UV-vis spectrophotometer (PE-950, PerkinElmer, USA).

### 4.2. Experimental Design

Twelve independent aquarium tanks were constructed to evaluate the antifouling and anti-biofilm properties of TRP (Figure 9). The tank volume was 50 L, and the seawater loading volume was 40 L. Half of the seawater was replaced with natural seawater obtained from Daya Bay of the South China Sea (22°38′17.54″ N, 114°05′52.35″ E) every 3 days. The 904 L stainless-steel slices (length × width × depth = 230 mm × 210 mm× 1 mm) that are commonly used in the marine environment were used as the adhering substance for biofilm formation; they were washed with 75% ethanol before use. As TRP is a photosensitive material, a VL-emitting diode irradiation source was mounted above the tanks to simulate sunlight irradiation. The temperature was maintained at 28 ± 1 °C, and the light intensity was kept at 8000 LUX with a 12 h/12 h light/dark cycle. The container water levels were adjusted and compensated daily with sterile distilled water.

There were four groups in total: the test group (PL; with TRP and light), control group P (with TRP only), control group L (with light only), and blank group N (with no TRP and no light). Seawater samples were used as the background reference. The TRP was evenly fixed on the filter membrane (acetate cellulose, Shanghai Xinya Purification Equipment Co., Ltd., China) and stuck onto the bottom of the tanks. The final concentration of TRP in each tank was 100 mg/L. Experiments were performed with three replicates for each treatment. The slices were immersed in seawater, and each tank was set up in triplicate. A schematic diagram of the experimental design and culture system are shown in Figure 9.

Based on the time required for the natural maturation of biofilms, the behavior of marine biofilms was continuously monitored for 4 weeks. To assess the biofilm communities and their functions in different time periods, samples were incubated under the above-mentioned conditions and collected at the first, second, and fourth weeks. The biofilm samples on the slices were washed gently three times with sterile seawater to remove unbound materials and collected for subsequent fluorescence observation and DNA extraction.

### 4.3. Environmental Parameters

The common physicochemical factors of seawater, including temperature and salinity, were measured in situ with an EXO1 Multiparameter Sonde (EXO1, YSI, USA). The pH value was measured using FiveEasy Plus™ (Mettler Teledo, Zurich, Switzerland). Chlorophyll a was measured using a Chlorophyll Fluorescence System (Phyto-PAM, Walz, Germany). Phosphate phosphorus (PO_4_^3−^), nitrate nitrogen (NO_3_^−^), nitrite nitrogen (NO_2_^−^), and ammonium nitrogen (NH_4_^+^) were measured using a Discrete Chemistry Analyzer (CleverChem Anna, DeChem-Tech, Germany). TOC was measured using a Shimadzu TOC-VCPH analyzer (Japan).

### 4.4. Biofilm Profiles

Fluorescence staining combined with CLSM was used to visualize the changes in microbial communities during the formation of biofilms. The biofilms adhering to the stainless-steel slices during different time periods were collected using a sterile knife. The treated biofilms were tested using a LIVE/DEAD^®^ BacLight™ Bacterial Viability Kit by following the manufacturer’s instruction. An anti-quenching reagent ProLong™ Gold was used after fluorescence staining. Different light path combinations for CLSM were chosen to observe the survival ratios of bacteria and the structure of the biofilm.

Quantification of biofilm parameters was processed with the COMSTAT software using the CLSM images [76]. Of the available parameters, we selected total biomass, average thickness, and roughness coefficient to evaluate the biofilms [77]. The abundances of live and dead biomass were evaluated based on the fluorescence intensity. An NIS-Elements Viewer (Nikon, Japan) was used to produce three-dimensional (3-D) transmission-fluorescence images of biofilms. Optical sections were 1 µm apart on the Z-axis. The length and width of the 3-D box were both 118 µm and the thickness was 25 µm.

### 4.5. DNA Extraction and Amplification

The incubated biofilm samples were washed gently three times with sterile seawater to remove loosely attached cells and then placed in aseptic centrifuge tubes. Triplicate biofilm samples were collected from three independent slices and filtered through 0.22 µm sterile cellulose-ester filter membranes (Merck Millipore, Billerica, USA) to concentrate the microbial biomass. DNA was extracted, quantified, and stored at −80 °C for further analysis.

Next-generation amplicon sequencing of 16S rRNA was conducted to analyze the biofilm bacterial communities. The V4-V5 variable region (515 forward primer: 5′-GTGCCAGCMGCCGCGGTAA-3′; 907 reverse primer: 5′-CCGTCAATTCMTTTRAGT-3′) of prokaryotic 16S rRNA genes was used [78]. The PCR reaction system contained 1 μL forward primer (10 μM), 1 μL reverse primer (10 μM), 3 μL template genomic DNA (20 ng/μL), 25 μL 2×Premix Taq (Shenggong Biotechnology, Shanghai Co. Ltd., Shanghai, China), and 20 μL nuclease-free water. The PCR program was as follows: initialization at 94 °C for 5 min, 30 cycles of denaturation at 94 °C for 30 s, annealing at 52 °C for 30 s, extension at 72 °C for 30 s, and finally elongation at 72 °C for 10 min. The PCR products were purified using an E.Z.N.A.^®^ Gel Extraction Kit (Omega, USA), followed by library construction and sequencing with the Illumina HiSeq 2500 platform.

### 4.6. 16S rRNA Gene Sequencing and Analyses

The 16S rRNA sequencing was performed using QIIME (version 1.9.1) [26], and the results were analyzed using Mothur version 1.35.1 [79]. OTUs were identified using a similarity threshold of 97% [80]. The taxonomy of each OTU was determined by RDP Classifier at an 80% threshold according to the Silva 16S rRNA database [81,82].

Taxonomic community structures were characterized based on 16S rRNA gene amplicons [31]. LEfSe was used to find marker species with different classification levels for different biofilms and seawater [83]. Rarefaction curves were constructed and alpha-diversity analyses (Simpson diversity, Shannon diversity, Chao 1 diversity, and observed OTUs) were performed using usearch-alpha_div. Beta-diversity was calculated using NMDS based on the Bray–Curtis dissimilarity matrix [84]. Furthermore, PERMANOVA was used to assess the influence of different incubation environments on the community variance [85]. One-way analysis of variance was used to determine the statistical significance of differences (*p* < 0.05) between all samples. In addition, the relationship of bacterial community composition with environmental factors was evaluated by RDA with CANOCO 5 [86].

An interaction network was established to identify co-occurrence patterns among the biofilm microbes [87]. To explore the differences of key species in different groups, we mixed data from the same group at different time points. To simplify the network, only the top 50 genera with higher relative abundance were determined. The co-occurrence network was analyzed through calculation of pairwise Spearman’s rank correlations based on the relative abundance of genus. Modular analyses and network visualization were carried out using Gephi 0.9.2. Topological properties, including average path length (the average number of steps along the shortest paths for all possible pairs of network nodes), node degree (the number of adjacent edges), betweenness centrality (the number of shortest paths going through a node), and modularity (a measure of the strength of division of a network into modules) were analyzed [50]. Nodes with low betweenness centrality and high degree were identified as keystone genus [28].

### 4.7. Metagenomic Sequencing and Analyses

Among the samples collected at four time points during the experiments, we selected eight representatives for metagenomic analysis, including the PL, P, L, and N groups in the first and second weeks. DNA extraction and purification were performed as described above. Considering the similar conditions for bacterial communities in replicate samples at the two timepoints, three replicates at the same timepoint were mixed into one sample for metagenomic sequencing.

Metagenomic sequencing was performed at Magigene (Guangzhou, China) on an Illumina NovaSeq 6000 platform. Raw data processing was carried out to obtain clean data for further analysis using Trimmomatic (v.0.36) [88]. Metagenomes of the biofilm replicates per sampling time were assembled using MEGAHIT (v1.0.6) [89,90]. To obtain preliminary stitching results (scaffolds), clean data analysis after quality control was conducted with de novo stitch [91]. As scaffolds may contain chimera sequences, we strictly broke scaffolds from one or more consecutive N positions to obtain fragments without N (called scaftigs) and filter out fragments below 500 bp [92]. In order to obtain low-abundance species sequences, unused reads were mixed and assembled [93]. The open reading frames of scaftigs (≥500 bp) generated by both mixed and single assembly were all predicted by MetaGeneMark (v3.38) [94]. Open reading frames with a length of less than 90 bp were filtered out from the prediction results. Redundancy was removed, and a unique initial gene catalogue was obtained using CD-HIT (v4.7), which was clustered by identity 95%, coverage 90%, and the longest sequences were chosen to be the representative sequences. Clean data for each sample were mapped to the initial gene catalogue using BBMAP software, and the number of reads to which genes mapped in each sample was obtained. The DIAMOND software was used to blast unigenes to the KEGG database [95]. The gene number of each sample in each taxonomy hierarchy was obtained based on the functional annotation and gene abundance results.

### 4.8. Statistical Analysis

*p* < 0.05 was considered statistically significant, and *p* < 0.01 was considered highly significant. The methods used for 16S rRNA diversity and metagenomic analysis are described in Section 2.6 and Section 2.7. In addition, structural equation models (SEMs) analysis was conducted by SPSSAU using an online tool (https://spssau.com, accessed on 5 October 2021).

## 5. Conclusions

The current study investigated the antifouling performance of TRP in natural seawater during the early stage of biofilm formation. Material and chemical structure analysis showed that VL-driven photocatalytic disinfection was the main contributor to the anti-biofilm capability of TRP. Reads-based analysis revealed significant difference in microbial communities among the TRP treatment group, control groups, and adjacent seawater group in terms of biodiversity, taxonomic composition, and microbial network structure. Metagenomic analysis further demonstrated that TRP affected the relative abundances of genes involved in EPS synthesis, QS, chemotaxis, biofilm formation, and antioxidant ability (Figure 10, schematic representation of mechanism of action). Taken together, these results indicate that TRP has an irreplaceable role in inhibiting the formation of biofilms, and that it is an environmentally friendly and photosensitive antifouling candidate. Development of photocatalysts from biological materials provides a green strategy for applications in marine biofouling. Although the present study has identified the ability of TRP to reduce marine biofouling in a laboratory setting, further investigations involving other fouling organisms in actual seawater are needed to establish the high-performance fouling-resistance efficacy of TRP for practical applications and development into a commercial product.

## Figures and Tables

**Figure 1 ijms-23-03837-f001:**
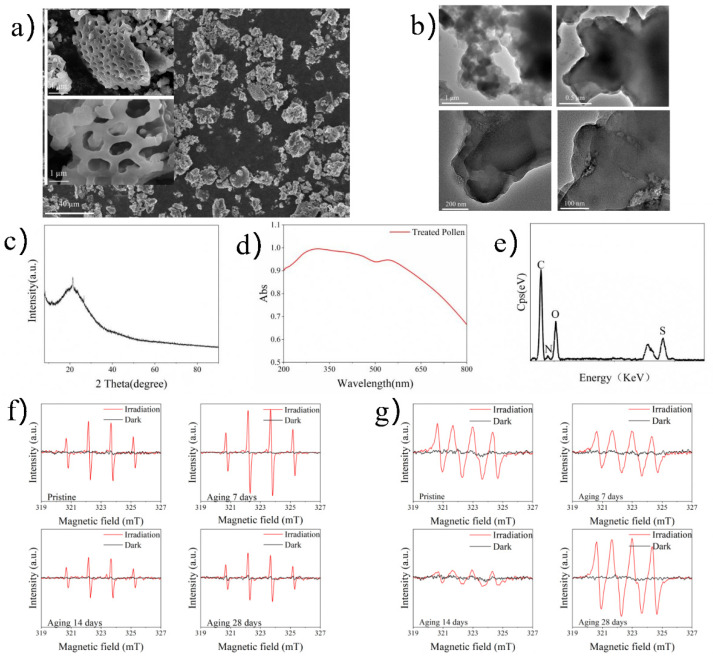
Characterization of TRP. (**a**) SEM images, (**b**) TEM images, (**c**) XRD pattern, (**d**) UV-vis DRS, (**e**) EDX characteristics, (**f**) hydroxyl radicals, and (**g**) superoxide radicals of TRP.

**Figure 2 ijms-23-03837-f002:**
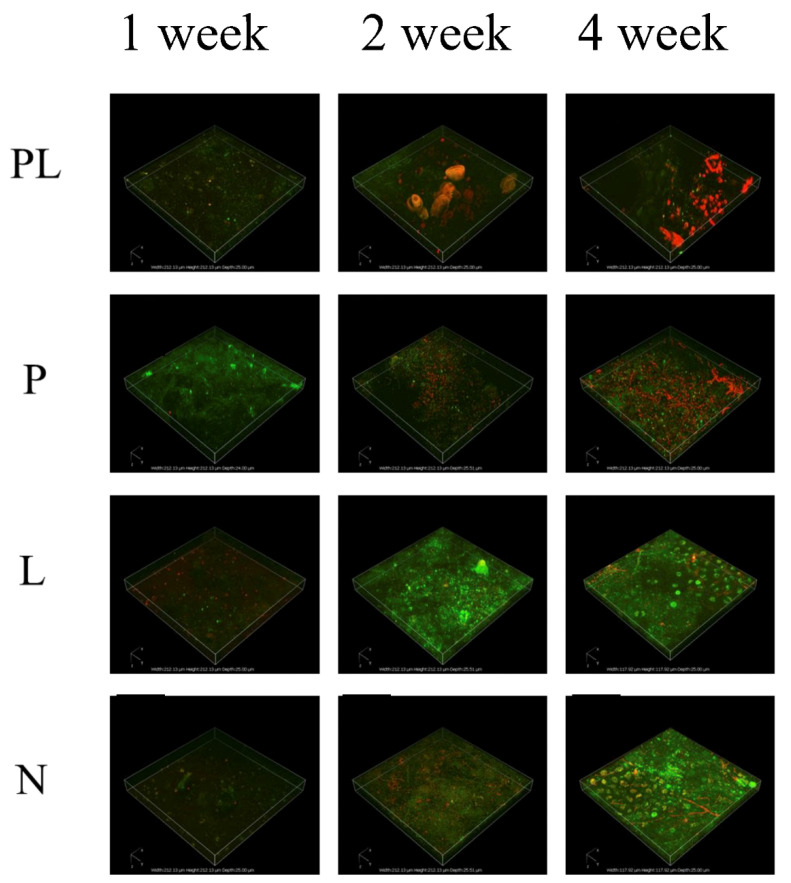
CLSM images of bacteria adhered to surfaces. Biofilms were simultaneously stained with propidium iodide (green) and SYTO^®^ 9 (red). The length and width of the 3-D box were both 118 µm, and the thickness was 25 µm.

**Figure 3 ijms-23-03837-f003:**
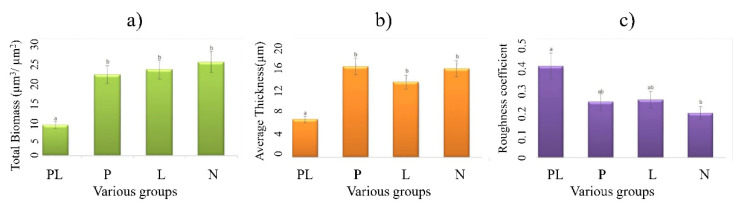
Quantification of biofilm formation of natural surface (taking second week as an example) using COMSTAT software, including (**a**) bio-volume, (**b**) average thickness, and (**c**) roughness coefficient. Error bars indicate SD (*n* = 3). Different letters indicate a statistically significant difference (*p* < 0.05) among the various groups.

**Figure 4 ijms-23-03837-f004:**
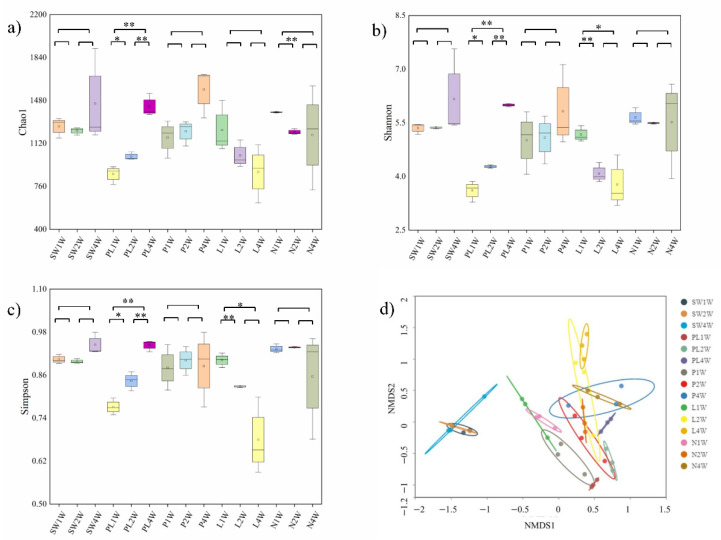
(**a**) Chao 1 index, (**b**) Shannon index, (**c**) Simpson index, and (**d**) NMDS based on the Bray–Curtis dissimilarities of bacterial communities. * *p* < 0.05, ** *p* < 0.01.

**Figure 5 ijms-23-03837-f005:**
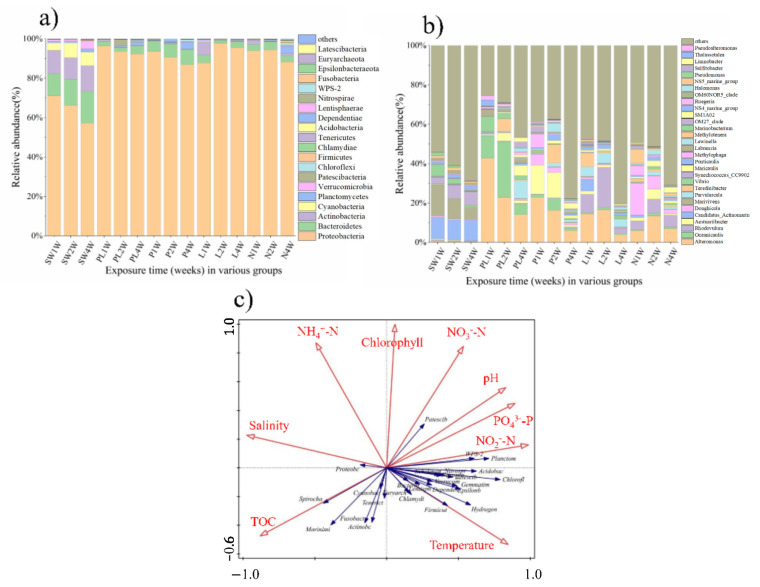
Relative abundance of different samples at (**a**) phylum and (**b**) genus levels, and (**c**) RDA of the correlations between bacterial community composition and environmental factors.

**Figure 6 ijms-23-03837-f006:**
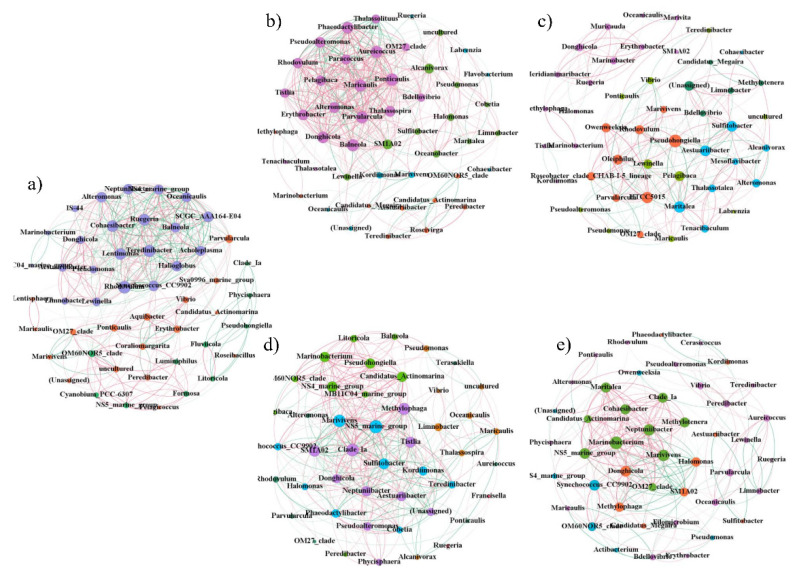
Network visualization of genus–genus interactions in (**a**) seawater group, (**b**) PL group, (**c**) P group, (**d**) L group, and (**e**) N group. Positive and negative correlations are shown in red and green, respectively. The nodes are colored according to different types of modularity classes. The size of each node is proportional to the node degree (Spearman’s |r| > 0.7, *p* < 0.05, top 50 most abundant genes).

**Figure 7 ijms-23-03837-f007:**
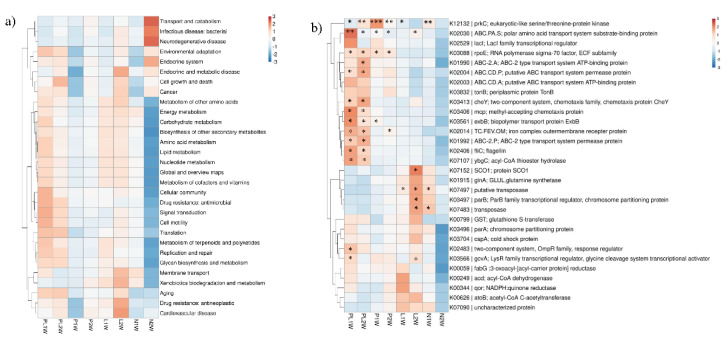
Distribution of (**a**) major functional pathways (level 2) and (**b**) functional genes in four biofilms (data were standardized and centralized). * two-fold, ** four-fold, *** eight-fold.

**Figure 8 ijms-23-03837-f008:**
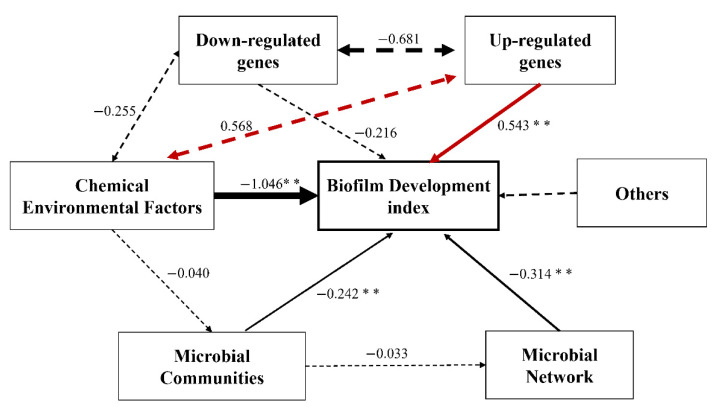
Structural equation modeling (SEM) showing the linkage among environmental factors, genes, species diversity, network factors, and biofilm factors. Dotted arrows indicate non-significant paths (*p* > 0.05). Red and black arrows indicate positive and negative relationships, respectively. The path widths are scaled proportionally to the path coefficient. ** *p* < 0.01.

**Figure 9 ijms-23-03837-f009:**
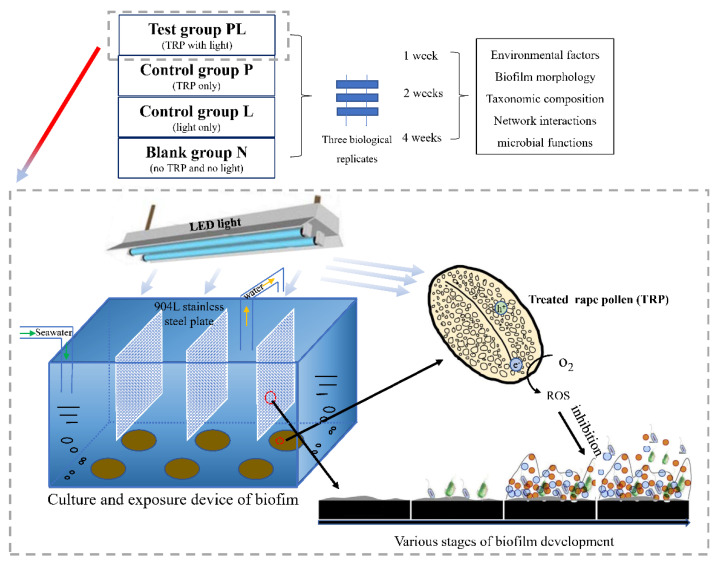
Culture and exposure of biofilm.

**Figure 10 ijms-23-03837-f010:**
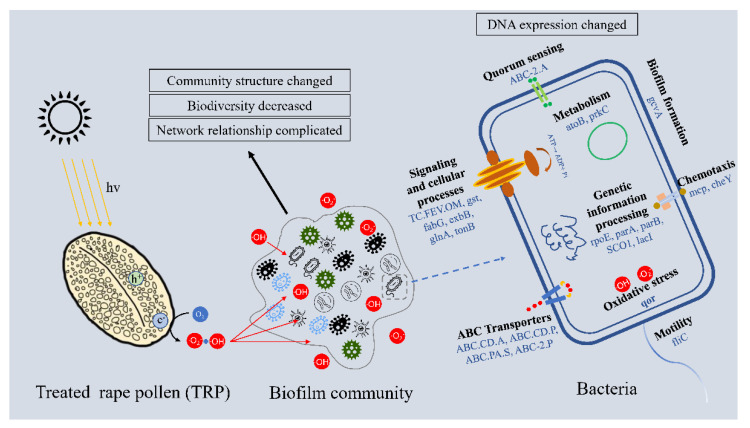
Schematic representation of mechanism of action.

**Table 1 ijms-23-03837-t001:** Topological properties of genus–genus interactions.

Sample	Number of Nodes	Number of Edges	Positive Correlation	Negative Correlation	Average Degree	Average Path Length	Average Clustering Coefficient	Figure Density	Modularity Index
Group A (PL)	48	488	70.90%	20.10%	10.167	2.835	0.649	0.216	0.281
Group B (P)	47	270	66.67%	33.33%	5.745	3.816	0.547	0.125	0.468
Group C (L)	46	372	62.37%	37.63%	8.087	2.456	0.525	0.180	0.346
Group D (N)	45	274	70.07%	29.93%	6.089	3.201	0.52	0.138	0.436
Seawater	49	442	56.11%	43.89%	9.02	2.934	0.554	0.188	0.398

## Data Availability

The metagenomic and 16S rRNA gene amplicon datasets have been deposited into the NCBI database with BioProject accession numbers PRJNA752478 and PRJNA751278, respectively.

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
