# Peer review of "The Microbial Mechanisms of a Novel Photosensitive Material (Treated Rape Pollen) in Anti-Biofilm Process under Marine Environment"

_ijms, 2022, doi:10.3390/ijms23073837_

Round 1

Reviewer 1 Report

Comments File is attached

Author Response

Please the attached file.

Reviewer 2 Report

The paper by Li et al. reports the physicochemical properties and antimicrobial mechanism of the photosensitive TRP. The paper provides some convincing evidence of photocatalytic disinfection of a steel surface under a marine environmental setting. The paper is interesting, methodology was appropriate, and the results and conclusions are convincing. However, some concerns pointed out below by the reviewer need to be addressed before it can be considered for publication.

  1. Typographical and grammatical errors can be seen throughout the paper. The “Results”, “Discussion” and “Conclusion” sections are comprehensible, although some parts are hard to follow because of grammatical errors. However, the “Introduction” and “Materials and Methods” sections are difficult to follow and need to be checked to improve the accuracy of English language used. For example: line 34 to 36, “Biofilms are high-profile in world-wide marine events…”. What do the authors mean by this sentence?
  2. The title is inappropriate and should be replaced with a more appropriate one. What do the authors mean by “….microbial mechanisms of a novel photosensitive material …”? Did the authors mean “antimicrobial mechanisms of a novel photosensitive material”? The title can perhaps be revised to: “Antimicrobial mechanisms of a novel photosensitive material (treated rape pollen) under marine environmental conditions”. Or, a more appropriate title should be provided.
  3. Throughout the paper, the terms “anti-biofouling”, “anti-fouling”, “antifouling” were used to refer to the same phenomenon. Using different terms to refer to the same phenomenon just adds to confusion. The authors should use the same term to refer the same phenomenon. In this paper, the authors should just use the term “antifouling”.
  4. 1, 2, 3, 4, 5 and 7: Tables and Figures should be self-explanatory and should be comprehensible as is without having to refer to the text for detailed description. For example, Fig.1 a and b: what do the red arrows mean? Fig. 2: what are PL, P, L and N? what are (A1), (A2), etc.?
  5. Materials and Methods lines 457 to 464:    If (RP) was completely dissolved in ethanol, what was washed in DI? Was TRP collected on the 0.45μm filter? Description is difficult to understand and hence should be revised.
  6. Line 383: “The possible microbial mechanisms of TRP in anti-biofilm process” What do the authors mean by this sub-tittle? Revise this sub-title.

Finally, I suggest that the authors have their paper checked by a native English speaker.

Reviewer 3 Report

This is a noteworthy research paper which analyses the antifouling potential of treated rape pollen (TRP), a material with recognized photosensitive properties. The authors were able to convincingly suggest that TRP can constitute a novel “green” antifouling material, which can influence negatively the early steps of biofilm formation with the presence of light irradiation. This study constitutes a more ingenious and more multidisciplinary approach then previous research which tends to be focused more exclusively on specific antifouling properties. The manuscript as a whole, is pertinent, well written, and follows correctly all methodology associated to anti-biofilm and biofilm characterization research. Specifically, the authors employed confocal laser scanning microscopy to visualize biofilm formation, as well a 16S rRNA and metagenomic sequencing analysis to scrutinize the bacterial communities and microbial functions. The environmental parameters of the study were also taken in consideration. This followed all the standard and latest protocols and allowed the authors to support their hypothesis, in particular the possible relation between environmental factors, genes, and diversity of the biofilm communities. Additionally, tables, the figures and particularly the concept illustrations highlight the main findings of this work and the conclusions are well grounded and based on the obtained results. The manuscript quotes the appropriate past references, albeit in some instances some more references could be added, and this allows to infer correctly the consequences of this work in the antifouling/antibiofilm domain. The level of written English is also appropriate, and the objectives and pertinence of this study are made clear to the reader. As whole, the manuscript does not have any major weakness, just the occasional mistake or lack of a few details. These comments are included in the attached list of comments.

Author Response

Please see the attacted file.
